# The Effect of Exercise Training on Myocardial and Skeletal Muscle Metabolism by MR Spectroscopy in Rats with Heart Failure

**DOI:** 10.3390/metabo9030053

**Published:** 2019-03-19

**Authors:** Mingshu Shi, Øyvind Ellingsen, Tone Frost Bathen, Morten A. Høydal, Tomas Stølen, Morteza Esmaeili

**Affiliations:** 1Department of Circulation and Medical Imaging, Norwegian University of Science and Technology, NO-7491 Trondheim, Norway; mingshu.shi@ntnu.no (M.S.); tone.f.bathen@ntnu.no (T.F.B.); morten.hoydal@ntnu.no (M.A.H.); tomas.stolen@ntnu.no (T.S.); mor.esmaeili@gmail.com (M.E.); 2Clinic of Cardiology, St Olavs Hospital, NO-7491 Trondheim, Norway; 3Clinic of Cardiothoracic Surgery, St Olavs Hospital, NO-7491 Trondheim, Norway

**Keywords:** metabolomics, MRS, magnetic resonance spectroscopy, cardiac metabolism

## Abstract

The metabolism and performance of myocardial and skeletal muscle are impaired in heart failure (HF) patients. Exercise training improves the performance and benefits the quality of life in HF patients. The purpose of the present study was to determine the metabolic profiles in myocardial and skeletal muscle in HF and exercise training using MRS, and thus to identify targets for clinical MRS in vivo. After surgically establishing HF in rats, we randomized the rats to exercise training programs of different intensities. After the final training session, rats were sacrificed and tissues from the myocardial and skeletal muscle were extracted. Magnetic resonance spectra were acquired from these extracts, and principal component and metabolic enrichment analysis were used to assess the differences in metabolic profiles. The results indicated that HF affected myocardial metabolism by changing multiple metabolites, whereas it had a limited effect on skeletal muscle metabolism. Moreover, exercise training mainly altered the metabolite distribution in skeletal muscle, indicating regulation of metabolic pathways of taurine and hypotaurine metabolism and carnitine synthesis.

## 1. Introduction

Heart failure (HF) is a pathophysiologic condition in which the heart is unable to pump blood at an adequate rate to meet the normal circulatory demand [1]. Community-based studies show that about 30% of patients die from HF within one year after receiving the diagnosis [2,3]. Despite the significant improvement in prognosis with current therapies, longevity and quality of life are markedly reduced [4].

HF is associated with reduced cardiac energy metabolism, and the imbalance between energy consumption and production has been referred to as “energy starvation” [5,6]. The impaired energy metabolism has many aspects, among which the most important relate to substrate utilization, oxidative phosphorylation, and adenosine triphosphate (ATP) transfer and utilization [7].

Decreased exercise capacity is a common symptom in HF patients, limiting their daily activities and reducing their quality of life [8,9]. Skeletal muscle dysfunction is reported in HF patients and includes impaired contractile function, muscle atrophy, fiber-type transition, and mitochondrial dysfunction [10,11,12]. Skeletal muscle atrophy is characterized by decreased fiber size, which is closely associated with decreased exercise capacity in HF [11]. The muscle fiber type ratio is shifted towards more type II fiber [13]. Similar to the failing heart, mitochondrial dysfunction is also commonly observed in the skeletal muscle of HF patients [14].

At the metabolic level, these functional changes in skeletal muscle during HF may be associated with many different pathways. For example, muscle atrophy can result from an imbalance between protein synthesis and protein degradation, which also regulates amino acid levels in the skeletal muscle [10,11,12]. Mitochondrial dysfunction, on the other hand, can be strongly associated with decreased glucose and fatty acid oxidation, which is suggested by the greater reliance on anaerobic metabolism [14]. These metabolic and functional changes in skeletal muscle, together with a decline of cardiopulmonary function, lead to decreased exercise capacity in HF. However, the detailed metabolic profile in skeletal muscle and the link between changes in skeletal muscle metabolism and cardiac metabolism need further investigation.

Despite interplay between several factors such as inborn genotype [15] and aging [16], exercise capacity can be improved by exercise training in both healthy individuals [17,18] and HF patients [19]. It has been demonstrated that lifelong exercise training can prevent aging-related cardiac structural modeling and thus reduce the risk of heart failure [20]. At the metabolic level, exercise training attenuates some of the skeletal muscle abnormalities by improving energy production [10,21]. In addition, exercise training can also affect cardiac function through the adrenergic system. It has been reported that exercise training ameliorates β adrenergic receptor (βAR) responsiveness in elderly subjects, thus contributing to a clinical improvement in cardiovascular health [22,23]. Although these studies extensively investigated the mechanisms of how exercise training benefits HF patients, detailed metabolic profile changes in myocardial and skeletal muscle in response to exercise training are still largely unknown. By detecting metabolites that are involved in these identified metabolic pathways, this study might provide useful information in clinical applications, such as treatment monitoring and biomarker identification.

To get a better understanding of the interplay between HF, exercise training, and the metabolic profile from myocardial and skeletal muscle, we performed moderate- and high-intensity exercise training in rats after surgically induced myocardial infarction to (i) determine the resultant changes in the metabolic profile of myocardial tissue and skeletal muscle and (ii) explore how these changes reflect the changes in metabolic pathways in both cardiac and skeletal muscle. We hypothesized that myocardial infarction and exercise training would lead to metabolic changes in cardiac as well as skeletal muscle. The observed changes differed between heart and skeletal muscle, and thus different metabolic pathways might be affected.

## 2. Results

### 2.1. Metabolic Profiles of Myocardial Tissue

We identified 14 metabolites in myocardial tissue (Table 1 and Appendix A), which were further used as input for principal component analysis (PCA). Myocardial metabolism differed in HF rats compared to Sham, independent of exercise (Figure 1A). The loading plot illustrated that metabolite levels of glucose, glycine, taurine, aspartate, succinate, and lactate were higher in HF groups, whereas levels of creatine, phosphocreatine, glycerophosphocholine, glutamine, glutamate, lysine, acetate, and alanine were lower (Figure 1B).

Two-way ANOVA confirmed the overall difference shown by PCA in myocardial metabolic profiles of HF and Sham and demonstrated that ten metabolites were significantly different (Table 1). Pairwise comparison showed that the level of glucose, phosphocreatine, creatine, taurine and phosphocholine in HF were significantly different from Sham at some exercise training intensities (Figure 2).

To identify the affected metabolic pathways among the groups including exercise intervention, data from the detected metabolites were analyzed using the MetaboAnalyst software. Significant differences in pathway regulation were only found between Sham-sed and HF-sed, among which the most important pathway was Arginine and Proline metabolism (Figure 3). No significant effect was observed after exercise training.

### 2.2. Metabolic Profiles of Skeletal Muscle

Using the same process as used for cardiac muscle above, 15 metabolites were identified from the MR spectra of skeletal muscle (Table 2 and Appendix A). According to the PCA score plot, no prominent metabolic differences were observed between the HF and Sham group (Figure 4A). However, ANOVA demonstrated that seven metabolites (glycine, taurine, acetate, lysine, alanine, lactate, and anserine) were significantly different, mainly as a result of exercise training (Table 2). Specifically, higher levels of anserine and lower levels of taurine were found to be associated with exercise training in Sham, whereas a lower level of lysine was found in HF-mod compared to HF-sed (Figure 5).

We also observed an interaction effect of HF and exercise training on the levels of glycine and taurine. Glycine was significantly lower, and taurine was significantly higher in HF-high compared to Sham-high. Alanine was the only metabolite influenced by HF, whereas the pair-wise comparison showed no difference regarding exercise.

In order to identify the affected pathways in skeletal muscle, the metabolite data was analyzed using MetaboAnalyst software (Figure 6). Although univariate analysis did not show any significant change of taurine level between HF-sed and Sham-sed (Figure 5), the enrichment analysis detected that both taurine and hypertaurine metabolism was altered between HF-sed and Sham-sed (Figure 6A). High-intensity exercise training significantly altered the pathways for taurine and hypertaurine metabolism in Sham, whereas moderate intensity training affected carnitine synthesis in HF (Figure 6B,C).

## 3. Discussion

The main finding of the present study was that the changes in the metabolic profile during HF mainly affected the heart, whereas exercise training mainly affected skeletal muscle. Conversely, HF had a limited effect on skeletal muscle metabolism, whereas exercise training had a limited effect on metabolism in the heart. Furthermore, different metabolic profiles were observed between heart and skeletal muscle. The main changes in myocardial tissue included creatine metabolism, aspartate metabolism, and glucose metabolism, whereas representative changes in skeletal muscle included taurine metabolism and carnitine metabolism. In addition, although both myocardial and skeletal muscle had altered amino acid metabolism, the specific amino acids type affected were distinct: glutamine was the main metabolite that changed in heart, whereas lysine level was mainly changed in skeletal muscle.

### 3.1. Metabolic Changes in Myocardial Tissue

#### 3.1.1. Creatine Metabolism

The most important metabolic change of HF in the myocardial tissue was the decline of creatine and phosphocreatine (PCr) (Figure 1 and Figure 2, Table 1). The reduced level of PCr suggests a reduction of the PCr/ATP ratio. Several studies have reported that decreased PCr/ATP ratio is associated with impaired myocardial energy metabolism in HF, which also reflects the change in creatine kinase activity and total creatine depletion [24,25]. The decreased PCr/ATP ratio is a known predictor of heart failure pathogenesis, and it is highly associated with cardiovascular mortality [26,27].

According to the enrichment analysis, the change in creatine and phosphocreatine levels was an integrated part of the arginine and proline metabolism (Figure 3). It has been reported that arginine and proline metabolism is linked to atherosclerosis and acute coronary syndrome in a rabbit model [28]. Thus, the differences in metabolism between HF and Sham observed in our study might share similar properties.

#### 3.1.2. Aspartate Metabolism

Aspartate, another metabolite involved in arginine and proline metabolism, increased significantly in HF (Figure 1 and Figure 2, Table 1). The biosynthesis of aspartate in vivo mainly comes from the transamination of other amino acids, such as glutamine and alanine [29]. The process of transamination can be facilitated by aminotransferase enzyme (ASAT). The ASAT level in serum is strongly associated with liver dysfunction followed by heart failure and has been considered as a biomarker for evaluating the severity of heart failure [30,31,32]. Therefore, the elevated aspartate in HF from our study might also signify an increased ASAT level post-infarction.

#### 3.1.3. Glutamine Metabolism

As an important amino acid for protein synthesis, as well as an energy substrate in dysfunctional cells, glutamine was significantly decreased in myocardial tissue from HF rats (Figure 1 and Figure 2, Table 1). Previous studies found that postischemic reperfusion of rat hearts with glutamine can cause full recovery of cardiac output and also significantly increase the myocardial ATP/ADP ratio, suggesting a potentially cardioprotective effect in rat hearts [33,34,35]. Whether glutamine also plays a specific role in chronic heart failure needs to be further explored.

#### 3.1.4. Glucose Metabolism

The increased myocardial glucose level in HF (Figure 1 and Figure 2, Table 1) suggests an elevated glucose uptake from blood to myocytes. Although no direct measurement of fatty acid was available, this finding might reflect a shift in substrate utilization from fatty acid to glucose in HF, as previously shown in many HF studies [36,37,38,39,40]. The shift in substrate utilization is regarded as the result of metabolic remodeling in failing hearts, as glucose provides greater energy production efficiency to compensate for the impaired energy metabolism [41,42].

Interestingly, the difference in glucose level only existed between the sedentary groups. When the HF rats were trained at high intensity, this difference was no longer evident (Figure 2). This result supports the notion of improved energy metabolism after exercise training in failing hearts with more ATP produced.

### 3.2. Metabolic Changes in Skeletal Muscle

Exercise training-induced several metabolic changes in skeletal muscle. In HF rats, the most marked changes as a result of exercise training were the lysine level (Figure 5, Table 2) and carnitine synthesis (Figure 6C).

In contrast to myocardial tissue, the metabolic profile of skeletal muscle shared more similarities between Sham and HF (Figure 4 and Figure 5). Although pairwise comparison indicated no significant difference in taurine level between HF-sed and Sham-sed, the enrichment analysis still showed that taurine and hypotaurine metabolism were altered between HF-sed and Sham-sed (Figure 6A). The exact reason for this contradiction is not very clear. A possible explanation might be the interaction effect between HF and training, as the enrichment results also showed the taurine and hypotaurine metabolism also differed at high-intensity exercise training only in Sham (Figure 6B).

#### 3.2.1. Taurine Metabolism

In Sham rats, we found reduced taurine levels when trained at high intensity (Figure 5 and Figure 6B, Table 2) [43,44]. Previous studies demonstrated taurine depletion in both slow-twitch and fast-twitch skeletal muscle after prolonged exercise in rodents. Changes in taurine level are reported to be associated with many physiological processes, including cellular oxidative stress, osmotic stress, and cell signaling [43,45]. Moreover, taurine depletion is associated with reduced SR calcium release, resulting in an impaired contractile function of skeletal and heart muscle [43,45]. Thus, chronic taurine supplementation has been suggested to increase intramuscular calcium concentration following muscle activation and muscle force improvement in rats [43,44].

Surprisingly, our findings that decreased taurine level following exercise training were only observed in Sham rats, but not in HF rats (Figure 5). Although detailed mechanisms are still not very clear, a possible explanation is that HF rats are more inactive and thus have lower taurine consumption.

#### 3.2.2. Lysine Metabolism

In HF rats, we found that moderate intensity training reduced skeletal muscle metabolism of lysine (Figure 5, Table 2). In previous studies, oral administration of lysine was found to help sustain muscle mass by preventing proteolysis in fasted rats [46]. Also, it has also been shown that supplementing lysine can significantly prevent muscle wasting by inhibiting the autophagic-lysosomal system [47]. These studies suggested that lysine plays an essential role in the synthesis and maintenance of skeletal muscle and that low levels of lysine were linked to reduced protein synthesis.

HF is associated with an imbalance of protein synthesis and degradation in skeletal muscle. Therefore, the effect of moderate training on lysine metabolism in HF rats in our study suggests beneficial effects, such as maintaining protein synthesis, improving homeostasis and thus preventing skeletal muscle wasting.

#### 3.2.3. Carnitine Metabolism

In HF rats, the enrichment analysis also suggested a link between glycine and lysine and carnitine synthesis (Figure 6C). Carnitine is an essential nutrient and plays a crucial role in many physiological processes, such as mitochondrial *β*-oxidation, and in the ubiquitin-proteasome system regulation [48,49]. The depletion of carnitine is associated with impaired function of long-chain fatty acid transfer across the inner mitochondrial membrane, which limits the subsequent *β*-oxidation [50]. In addition, it has been shown that the synthesis of carnitine is beneficial for the recovery of skeletal muscle damage, which can be used as a therapeutic target in skeletal muscle-associated disease [48].

Therefore, the enrichment analysis in our study suggests that it would be interesting to further characterize the changes in carnitine metabolism in skeletal muscle during HF, which is potentially useful for exploring new therapeutic targets.

### 3.3. Limitations

The rat model of HF is a major limitation regarding generalization to humans. Mass specific metabolic rate is higher in rats than humans, and thus, there is a lower capacity to maintain homeostasis [51]. Unlike HF patients, these rats were young and without comorbidities. The use of larger animal models raises major ethical, economic, and methodological concerns. On the other hand, HF rats can give insight into which metabolites are affected due to the HF and not due to comorbidities. The current study is also purely descriptive and not mechanistic in how metabolites affect cardiac and skeletal muscle functions. The number of animals in the study is low and limits the strength. However, the metabolic signatures in the different groups give clues for further studies on biomarkers in vivo and mechanistic studies.

## 4. Materials and Methods

### 4.1. Rat Heart Failure Model

All experiments were conducted according to the Guide for the Care and Use of Laboratory Animals published by the US National Institutes of Health (NIH Publication No. 85-23, revised 1996). The experiments were designed according to the guidelines from the Federation of European Laboratory Animal Science Associations (FELASA), EU animal research directive (86/609/EEC) and Council of Europe (ETS 123) and the EU directive (2010/63/EU) and approved by the national animal ethics commitee, 4283. The 3 R’s (replacement, reduction, and refinement) have specifically been addressed when designing the study.

Rats were anesthetized with 5% isoflurane, then intubated and ventilated with 1.5% isoflurane in a 30% O_2_/70% N_2_O mixture. HF was induced by ligation of the descending coronary artery, leading to large myocardial infarction (MI) in the left ventricle (LV), as previously described [52,53]. After left thoracotomy, the pericardium was opened, and in the MI group, the descending artery was ligated with a polyester suture (Ethibond 6-0, needle Rb-2, Ethicon; Norderstedt, Germany). Sham rats underwent the same surgical procedure, except for ligation of the descending coronary artery. Buprenorfin (0.04 mg/kg) was injected subcutaneously during the surgery and repeated 8 h thereafter to relieve pain. After four weeks, rats with MI operation were examined by echocardiography to determine the extent of MI. Only rats with an MI size of 40–50% of the left ventricle were included, which gave an ejection fraction of about 20% (data not shown).

### 4.2. Exercise Training and VO_2max_ Testing

Four weeks after MI (or Sham) surgery, VO_2max_ was measured on an incline treadmill (25°) in a metabolic chamber as previously described [54]. After the test, Sham and HF rats were randomized to 6 subgroups: Sham rats at high-intensity exercise (Sham-high), moderate exercise (Sham-mod), and sedentary control (Sham-sed), and corresponding HF groups (HF-high, HF-mod, and HF-sed). The exercise training was conducted for 60 min/day, five days/week, for six weeks. Both moderate and high-intensity exercise training started with 10 min warm-up at 50–60% of maximal oxygen uptake (VO_2max_). The 60 min high-intensity exercise included 10 cycles of 4 min running at 85–90% VO_2max_ and 2 min at 50% of VO_2max_. For moderate-intensity exercise training, rats ran for 4 min at 60–70% VO_2max_ separated with 2 min at 50% VO_2max_ for the same distance as their respective high-intensity group. Moderate-intensity groups started exercising for about 80 min and gradually increased training duration to approximately 110 min at the end of the training period to match the high-intensity group with regard to distance (and energy expenditure). At the start, during, and at the end of the training period, VO_2max_ was measured to ascertain and adjust the band speed to maintain the desired running intensity. The sedentary rats did not exercise, and VO_2max_ was measured before and after the training period.

### 4.3. Tissue Extraction

VO_2max_ was measured at least five days before sacrifice, and the last exercise session was performed 24 h prior to sacrifice and tissue harvesting. The rats were anesthetized, and the hearts and skeletal muscle were quickly removed and placed in ice-cold saline for dissection. Performing an identical surgical procedure and by snap-freezing, the time from removal of the heart or skeletal muscle to snap freezing was approximately 1 min and did not differ between groups. The skeletal muscle extracted was soleus muscle. Metabolites were extracted from myocardial tissue samples by using perchloric acid, as previously described [55], and from skeletal muscle samples by using a modified dual phase extraction protocol [56]. After extraction, 48 samples from skeletal muscle and 30 samples from myocardial tissue were frozen at −80 °C, lyophilized, and stored at 4 °C until MR analysis.

### 4.4. Proton Magnetic Resonance Spectroscopy (MRS)

Before MRS analysis, samples were dissolved in deuterium oxide (D_2_O, Sigma-Aldrich Corporation, St. Louis, MO, USA). The pH of all samples was adjusted to the same level (pH ~ 7) by perchloric acid and potassium hydroxide. MRS was performed using a Bruker Avance III Ultra-shielded Plus 600 MHz spectrometer (Bruker Biospin GmbH, Rheinstetten, Germany) equipped with a 5 mm QCI Cryoprobe with integrated, cooled preamplifiers for ^1^H, ^2^H, and ^13^C. This MR system provided a fully-automated experiment equipped with a SampleJet in combination with Icon-NMR on TopSpin v3.1 software (Bruker Biospin). For samples from skeletal muscle, the MR spectra were obtained at 301.3 K using one-dimensional standard nuclear overhauser effect spectroscopy (1D-NOESY) (noesygppr1d; Bruker Biospin) with the following acquisition parameters: 128 scans, acquisition time of 2.73 s, relaxation delay of 4 s, free induction decay (FID) size of 65,536, and a spectral width of 20.0243 ppm. For samples from myocardial tissue, the MR spectra were obtained at 298.0 K using 1D standard Carr–Purcell–Meiboom–Gill (CPMG) pulse program (cpmgpr1d; Bruker Biospin) with the following acquisition parameters: 64 scans, acquisition time of 2.73 s, relaxation delay of 10 s, FID size of 65,536, and a spectral width of 20.0243 ppm. In CPMG MR spectra, signals from macromolecules are suppressed for further enhanced detection of low molecular weight metabolites.

### 4.5. Spectral Processing and Statistics

MR spectra were automatically Fourier transformed with an exponential line broadening of 0.3 Hz, phased, and baseline corrected in TopSpin. Preprocessed spectra were transferred into MATLAB R2013b (The Mathworks, Inc., Natick, MA, USA) and referenced to the Trimethylsilylpropanoic acid (TSP) peak at 0 ppm before peak alignment. Three Low-quality spectra with poor water suppression and poor shim were removed from further analyses. Chemical shift differences were corrected by the Icoshift algorithm [57]. Metabolites were assigned using NMR Suite 7.5 software (Chenomix Inc., Edmonton, AB, Canada). The area under the curve (AUC) of individual metabolite peaks was calculated using MATLAB. Prior to integration, the spectra were binned (bin size 0.01 ppm) and normalized by total area [58]. AUC of individual metabolites was used as input variables for multivariate analysis. Multivariate analysis was performed in MATLAB with PLS Toolbox 8.0.2 (Eigenvector Research Inc., Manson, WA, USA). After auto-scaling, PCA was performed to explore the metabolic profiles. Univariate analysis and visualization were performed in R version 3.4.1 and GraphPad Prism version 8.0.2 (GraphPad Software Inc., San Diego, CA, USA). Two-way ANOVA was used to assess the effects of heart failure and the different exercise intensities on the AUC of metabolites. Pairwise comparison within each group was performed using Tukey’s Post Hoc Test. Enrichment and pathway exploration analysis were performed using the interactive web-based application MetaboAnalyst 4.0 (http://www.metaboanalyst.ca/) [59]. To avoid the false positives from multiple testing, adjusted *p* value (or False Discovery Rate, FDR) was used to test for significant differences. The difference was considered significant if the FDR value was <0.05.

## 5. Conclusions

HF mainly affected the myocardium, whereas exercise training affected the skeletal muscles, and the distribution of metabolites in HF and exercise training were different in the myocardium compared to the skeletal muscle. However, the myocardial metabolic signature was not sensitive to exercise training.

## Figures and Tables

**Figure 1 metabolites-09-00053-f001:**
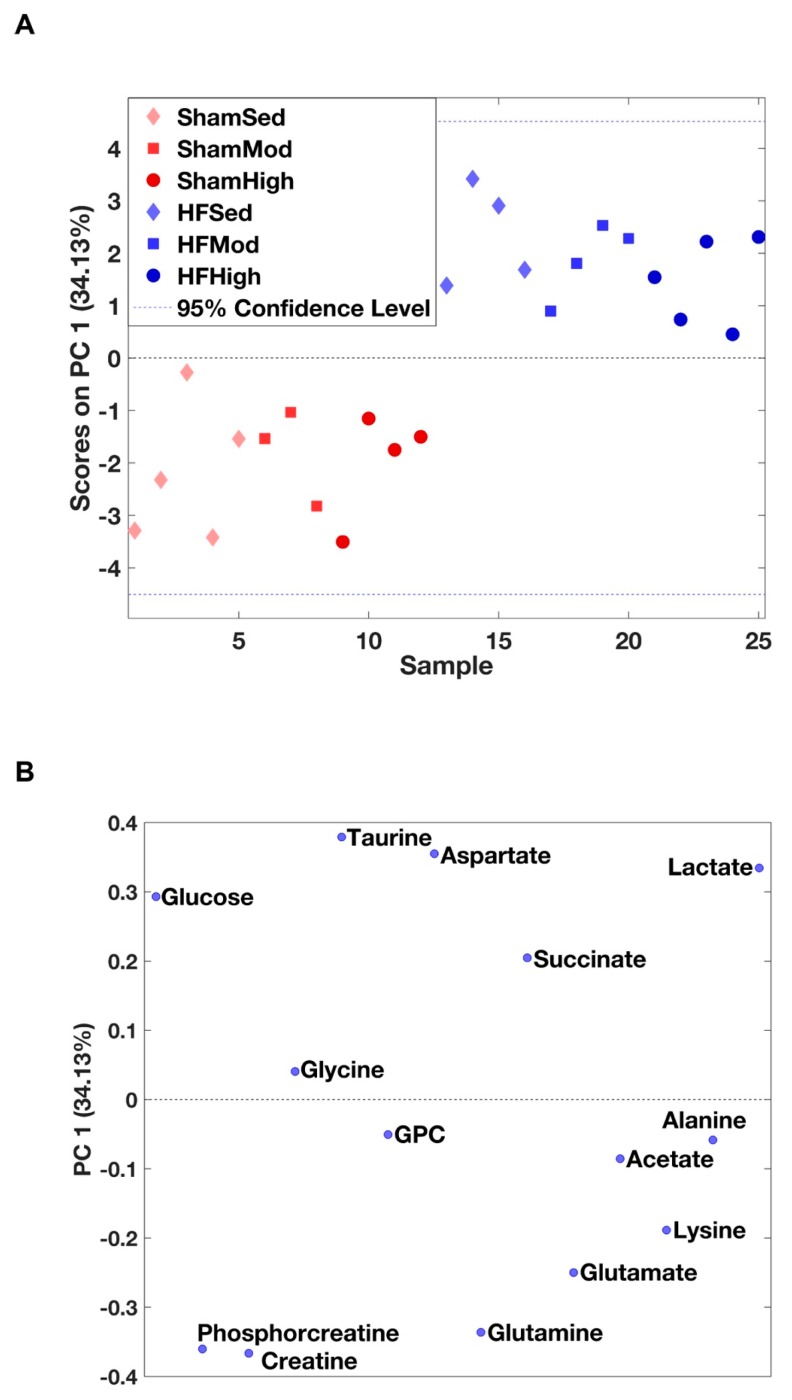
Principle component analysis (PCA) of myocardial metabolites. (**A**) The score plot from the PCA of the myocardial metabolites demonstrates a significant difference in the metabolite distribution between HF and Sham based on the first principal component (PC1), which explains 34% of the total variation. (**B**) The loading plot showing the contribution of the individual metabolites to the model.

**Figure 2 metabolites-09-00053-f002:**
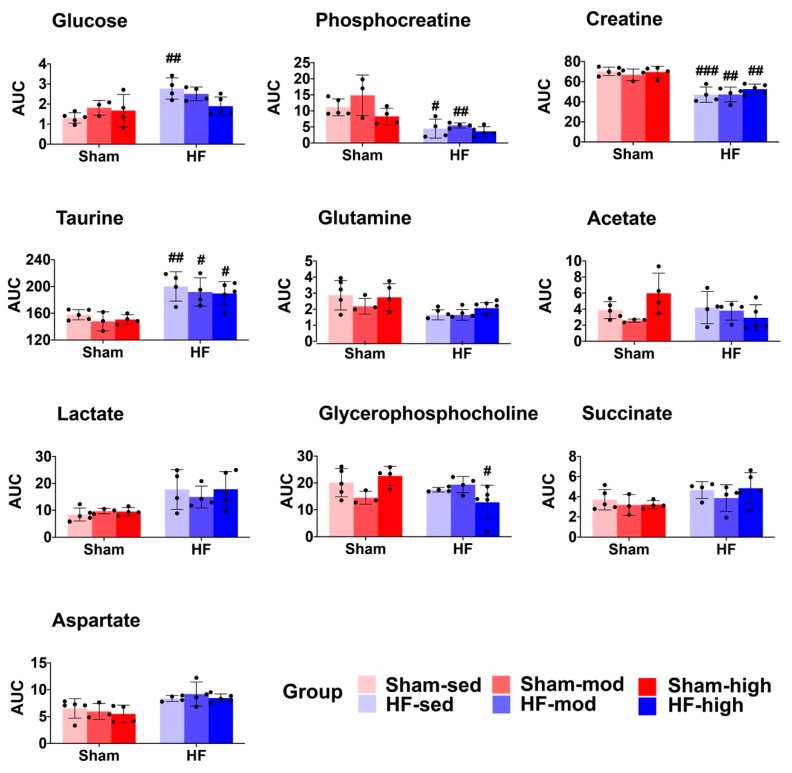
The scatter and bar plots show the relative metabolite levels (AUC) of the 10 metabolites that are significantly different in the ANOVA model (Table 1). Pairwise comparison within each group was performed using Tukey’s Post Hoc Test. In total 25 samples were distributed into 6 subgroups: Sham-sed, *n* = 5; Sham-mod, *n* = 3; Sham-high, *n* = 4; HF-sed, *n* = 4; HF-mod, *n* = 4; HF-high, *n* = 5. “#”above the HF bars donates that the exercise training subgroup was significantly different compared to its Sham counterparts (e.g., Sham-sed vs. HF-sed). #, ##, ### denote *p* values < 0.05, 0.01, and 0.001, respectively. Five metabolites (glutamine, acetate, lactate, succinate, and aspartate) were significantly different in the ANOVA model but showed no significant difference in the pairwise comparison.

**Figure 3 metabolites-09-00053-f003:**
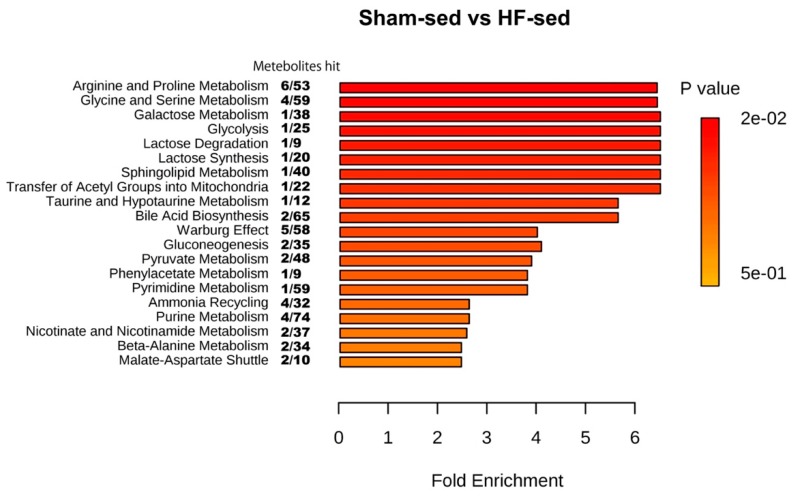
Enrichment analysis of the myocardial tissue. The pathways shown were significantly different between sham-sed and HF-sed, sorted by fold enrichment and *p* value. False Discovery Rate (FDR) > 0.05 was considered as statistically different.

**Figure 4 metabolites-09-00053-f004:**
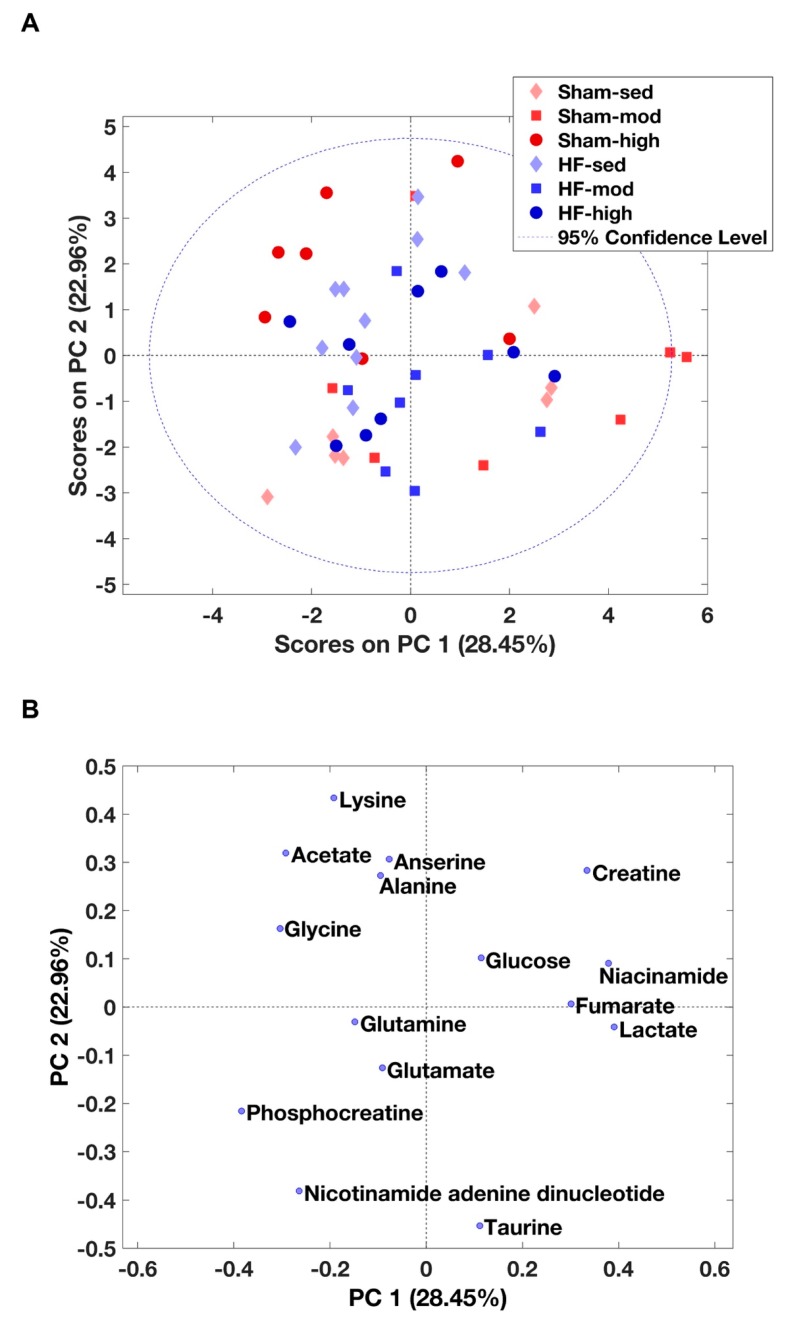
PCA of skeletal muscle metabolites. (**A**) The score plot from PCA within all skeletal muscle metabolites. Principal components 1 and 2 (PC1 and PC2) explain the model with 28% and 22% values, respectively. However, no obvious separation between HF and Sham was visually observed. (**B**) The contribution of the individual metabolites to the model.

**Figure 5 metabolites-09-00053-f005:**
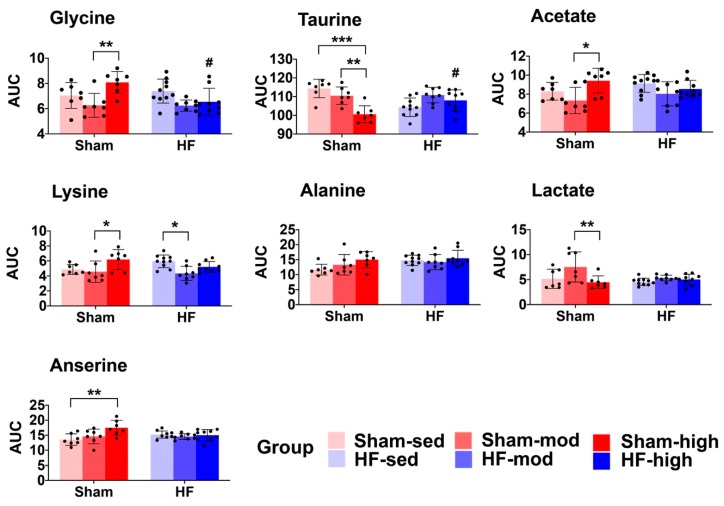
The skeletal muscle metabolic profile. The scatter and bar plots show relative metabolite levels (AUC) of the seven metabolites that are significantly different in the ANOVA model (Table 2). Pairwise comparison within each group was performed using Tukey’s Post Hoc Test. In total, 48 samples were distributed into 6 subgroups: Sham-sed, *n* = 7; Sham-mod, *n* = 7; Sham-high, *n* = 7; HF-sed, *n* = 10; HF-mod, *n* = 8; HF-high, *n* = 9. “#”above the HF bars denotes that the exercise training subgroup is significantly different (*p* < 0.05) compared to its Sham counterpart (e.g., Sham-sed vs. HF-sed). “*” denotes a significant difference between different exercise groups which underwent the same surgical process (e.g., Sham-sed vs. Sham-high). *, **, *** denotes *p* value < 0.05, 0.01, and 0.001, respectively. Although alanine was recognized as significantly different in the ANOVA model, the pairwise comparison under the exercise training or surgical factor did not result in a significant difference.

**Figure 6 metabolites-09-00053-f006:**
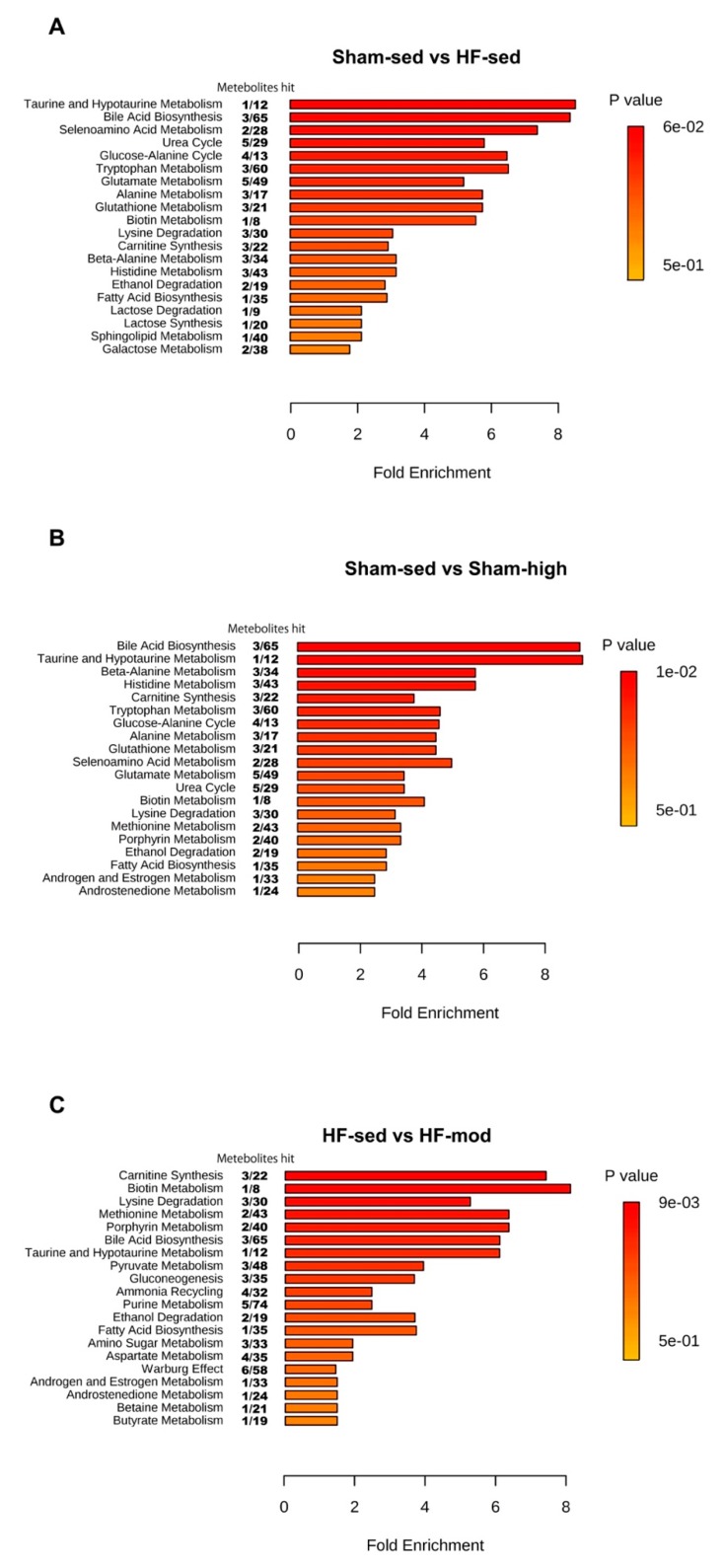
Enrichment analysis of skeletal muscle tissue. The pathways shown were significantly different between (**A**) Sham-sed and HF-sed; (**B**) Sham-sed and Sham-high; and (**C**) HF-sed and HF-mod, sorted by fold enrichment and *p* value. FDR > 0.05 was considered as statistically different.

**Table 1 metabolites-09-00053-t001:** ANOVA summary of myocardial tissue samples. In total, 14 metabolites were identified. The significance levels (*p* value) of two factors (surgery: Sham vs. heart failure (HF); exercise training intensities: Sedentary vs. moderate vs. high) are shown.

Metabolites	Surgery	Exercise	Interaction
Glucose	<0.001	-	0.046
Phosphocreatine	<0.001	0.050	-
Creatine	<0.001	-	-
Taurine	<0.001	-	-
Glutamine	0.0026	-	-
Acetate	-	-	0.037
Lactate	<0.001	-	-
Glycerophosphocholine	-	-	0.012
Succinate	0.026	-	-
Aspartate	<0.001	-	-
Glycine	-	-	-
Glutamate	-	-	-
Lysine	-	-	-
Alanine	-	-	-

**Table 2 metabolites-09-00053-t002:** ANOVA summary of the skeletal muscle sample. In total, 15 metabolites were identified. Significance levels (*p* value) of two factors (surgery: Sham vs. HF; exercise training intensities: Sedentary vs. moderate vs. high) are shown.

Metabolites	Surgery	Exercise	Interaction
Glycine	-	0.0044	0.014
Taurine	-	0.0056	<0.001
Acetate	-	0.0082	-
Lysine	-	0.0033	0.019
Alanine	0.043	-	-
Lactate	-	0.012	-
Anserine	-	0.032	0.010
Glucose	-	-	-
Phosphocreatine	-	-	-
Creatine	-	-	-
Glutamine	-	-	-
Glutamate	-	-	-
Fumarate	-	-	-
Niacinamide	-	-	-
NAD	-	-	-

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
