# Peer review of "The Effect of Exercise Training on Myocardial and Skeletal Muscle Metabolism by MR Spectroscopy in Rats with Heart Failure"

_metabolites, 2019, doi:10.3390/metabo9030053_

Round 1
Reviewer 1 Report
The authors investigated the effect of exercise training on metabolic profile of myocardial and skeletal muscle tissues in post-infract heart failure (HF) in rats. The global analysis for metabolites using proton magnetic resonance spectroscopy (MRS) revealed that 10 metabolites in the heart were different between sham and HF, whereas no significant difference in metabolites was observed in the skeletal muscle between two groups. In the myocardial tissue, exercise training had no effect on metabolic profile in sham as well as HF rats. In contrast, exercise training significantly altered taurine and hypertaurine metabolism in sham and carnitine synthesis in HF. This study is insightful at the point which they examined the impact of exercise training on metabolic profile in each of myocardial and skeletal muscle tissue in HF. Their findings suggest that exercise training exerts a benefit in HF by altering metabolism not in the heart but the skeletal muscle. However, due to the descriptive nature of this study, the pathophysiological role of exercise-induced metabolic patterns remains unknown. Specifically, it remains undetermined whether the metabolic patterns correlate a specific molecular change in skeletal muscle in HF. In addition, there are several issues on experimental methods and animal characteristics that should be addressed. Specific comments are listed below.
Major comments:
1) What is the pathophysiological role of exercise-induced change in metabolic profile in the skeletal muscle? If lysine and carnitine metabolism in skeletal muscle is involved in the form of pathological condition in HF, the association of those metabolites with molecular changes involved in muscle atrophy and fatty acid oxidation would add some information to address the issue.
2) The basic characteristics of sham and HF rats should be presented. How much were infarct size and the weights of heart and skeletal muscle? Please specify which part of skeletal muscles was excised and used for MRS analysis.
3) Please more extensively discuss the difference in the effect of exercise training on metabolic pathways between heart and skeletal muscle. What is the clinical relevance?
Minor comments:
Please clarify the number of rats used in the analysis in Figure 2 and 5.
Author Response
Point 1: What is the pathophysiological role of exercise-induced change in metabolic profile in the skeletal muscle? If lysine and carnitine metabolism in skeletal muscle is involved in the form of pathological condition in HF, the association of those metabolites with molecular changes involved in muscle atrophy and fatty acid oxidation would add some information to address the issue.
Response 1: We thank the reviewer for the suggestion. In the new version of manuscript, the physiological role of altered lysine and carnitine level has been expanded. One reference is added to support that lysine can prevent muscle wasting, by inhibiting the autophagic-lysosomal system (Page 12, Line 264). Another reference is added to describe the relationship between carnitine and long-chain fatty acid transfer (Page 13, Line 277). We hope that these newly added contents can better describe the exercise-induced change in metabolic profile in skeletal muscle.
Point 2: The basic characteristics of sham and HF rats should be presented. How much were infarct size and the weights of heart and skeletal muscle? Please specify which part of skeletal muscles was excised and used for MRS analysis.
Response 2: We agree that this is important to present. Unfortunately, we do not have weight of the heart and the soleus muscle. However, as stated in the manuscript all the MI rats included in the study had an estimated infarct size of 40-50% of the LV which gives EF down towards 20%. Below is a figure describing EF after the 8-week training period from the rats in the present study. This figure has been included in a different paper describing cardiac function and excitation contraction coupling extensively.
EF has been added to the Material and Methods section where we describe the animal model (Page 13, Line 312).
Representative picture of infarct size has been added to Figure S1: Example image of infarct size stained with hematoxylin-eosin and saffron.
Sample weights have now included in Table S1 and Table S2.
In the Material and Methods section, we have specified that soleus muscle was used in MRS analysis (Page 14, Line 335).
Point 3: Please more extensively discuss the difference in the effect of exercise training on metabolic pathways between heart and skeletal muscle. What is the clinical relevance?
Response 3: More detailed description about the different metabolic changes between heart and skeletal muscle have been added to the Discussion section (Page 11, Lines 185-194). Briefly, this difference comes from two aspects. Firstly, the main factors that change metabolic profile between myocardium and skeletal muscle are different: MI and HF mainly affected the myocardium, and exercise training mainly changed skeletal muscle metabolites. Secondly, the metabolites that are changed the most are different between heart and skeletal muscle: Changes in myocardium mainly include creatine metabolism, aspartate metabolism and glucose metabolism, whereas changes in skeletal muscle include taurine metabolism and carnitine metabolism.
The potential clinical relevance of this study might provide useful information in clinical applications such as treatment monitoring and biomarker identification of HF and is described in the introduction (Page 2, Lines 69-71).
Point 4: Please clarify the number of rats used in the analysis in Figure 2 and 5.
Response 4: We have now included the sample number in the figure legends of both Figure 2 (Page 5, Line 109) and Figure 5 (Page 9, Line 157).

Reviewer 2 Report
Although this reviewer warmly welcomes this manuscript, some questions should be addressed:
The rationale behind performing the study is not clearly presented to the Readers. A more integrated appraisal of the relevant literature would be appropriate to provide the context for the study.
The functional role of the adrenergic system in linking physical activity and cardiovascular health in heart failure (Iaccarino G et al. Front Physiol. 2013 Aug 12;4:209; Maturitas. 2016 Nov;93:65-72) should be discussed.
Data on cardiac function (EF, FS, HR, etc.) should be provided.
Representative pictures of the infarct size must be shown.
The strengths and limitations of the study should be deeply addressed, taking into account sources of potential bias or imprecision: Discuss both direction and magnitude of any potential bias.
Bar graphs with error bars do not allow direct evaluation of the distribution of the data. The authors should present their continuous data in scatter/dot plots (especially in case of a limited number of observations), showing the individual data points together with the average/error bars.
All legends should include specific "n" for each (and every) treatment group and a description of the statistics used for each experiment.
English language (syntax, grammar, correct choice of words, correct use of adjectives and adverbs) should be substantially improved throughout the text.
Author Response
Response to Reviewer 2 Comments
Point 1: The rationale behind performing the study is not clearly presented to the Readers. A more integrated appraisal of the relevant literature would be appropriate to provide the context for the study.
Response 1: Thanks to the reviewer for this comment. The Introduction section in the revised version has been updated with more literature that is relevant. We have expanded the content describing the relationship between physical activity and cardiovascular health. First, we introduce that HF is strongly associated with exercise capacity before mentioning factors influencing exercise capacity and then how exercise training can improve cardiovascular health and skeletal muscle function. Finally, we introduce the aim of this study: To better understand of the interplay between HF, exercise training the metabolic profile from myocardial and skeletal muscle, we performed moderate and high intensity exercise training in rats after surgically induced myocardial infarction to (i) determine the resultant changes in the metabolic profile of myocardial tissue and skeletal muscle; (ii) explore how these changes reflect the changes in metabolic pathways in both cardiac and skeletal muscle.
Point 2: The functional role of the adrenergic system in linking physical activity and cardiovascular health in heart failure (Iaccarino G et al. Front Physiol. 2013 Aug 12;4:209; Maturitas. 2016 Nov;93:65-72) should be discussed.
Response 2: Thank you very much for the suggestion. We agree that this is a very good aspect to describe the mechanism of the beneficial effects of exercise training on cardiovascular health. In the updated version, we included references that illustrate the functional role of adrenergic system in the heart during aging, and the impact of exercise training in β adrenergic receptor sensitivity (Page 2, Lines 64-66).
Point 3: Data on cardiac function (EF, FS, HR, etc.) should be provided.
Response 3: We agree that this is important data. This is a sub-study of a bigger study where we aimed to determine some of the exercise induced improvements of excitation contraction coupling and electrophysiology. As stated in the manuscript all the MI rats included in the study had an estimated infarct size of 40-50% of the LV which gives EF down towards 20%. Below is a figure describing EF after the 8-week training period.
EF has been added to the Material and Methods section where we describe the animal model (Page 13, Line 312).
Point 4: Representative pictures of the infarct size must be shown.
Response 4: Representative picture of infarct size has been added to Figure S1: Example image of infarct size stained with hematoxylin-eosin and saffron.
Point 5: The strengths and limitations of the study should be deeply addressed, taking into account sources of potential bias or imprecision: Discuss both direction and magnitude of any potential bias.
Response 5: We agree with the reviewer that the strength and limitation need to be further considered. In the current version, we have added a corresponding paragraph in the Discussion section (Page 13, Line 283).
Point 6: Bar graphs with error bars do not allow direct evaluation of the distribution of the data. The authors should present their continuous data in scatter/dot plots (especially in case of a limited number of observations), showing the individual data points together with the average/error bars.
Response 6: Thanks for the reviewer’s suggestion. We agree that the scatter plots are better illustrative regarding to the distribution of the individual data, especially for low sample size in our case. In the updated version of manuscript, we added the scatter points in Figure 2 and Figure 5. In order to keep the Figures format constant with previous ones, the bar plots with error bars are kept. They are now mixed plots with both scatter and bars.
Point 7: All legends should include specific "n" for each (and every) treatment group and a description of the statistics used for each experiment.
Response 7: Thanks to the reviewer for this suggestion. We have now included the sample number and statistics description in the figure legends of both Figure 2 (Page 5, Line 109) and Figure 5 (Page 9, Line 157).
Point 8: English language (syntax, grammar, correct choice of words, correct use of adjectives and adverbs) should be substantially improved throughout the text.
Response 8: In this new version, a native English speaker has revised the manuscript.

Round 2
Reviewer 1 Report
The authors of the present manuscript elegantly answered to all the different concerns raised by the reviewer. I have no further concern.
Author Response
Point 1: The authors of the present manuscript elegantly answered to all the different concerns raised by the reviewer. I have no further concern.
Response 1: We are glad to see that the reviewer is satisfied with current version and we really thank the reviewer for providing the suggestions before.

Reviewer 2 Report
There are 61 references mentioned in the text, but only 47 are listed.
Author Response
Point 1: There are 61 references mentioned in the text, but only 47 are listed.
Response 1: We would like to thank the reviewer for his comment on the reference section. We forgot to update the Endnote citation after revision. In the revised version of the manuscript, we have corrected the reference list. With the refresh Endnote update the total 59 references is now referred in the reference section of the manuscript.

Round 3
Reviewer 2 Report
-